# Literature Review of Mobility as a Service

**Benjamin Maas** 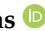

Faculty of Business and Economics, Technische Universität Dresden, Helmholtzstr. 10, 01069 Dresden, Germany; benjamin.maas@tu-dresden.de; Tel.: +49-176-820-32865

**Abstract:** The number of publications about mobility as a service (MaaS) has increased rapidly in the past years, spreading into various fields. In this paper, a total of 127 scientific publications about MaaS are reviewed and categorized into an overarching thematic framework in order to highlight key areas of research and further potential. Three research questions are highlighted in this review: (1) which topics are discussed in the existing MaaS literature? (2) what are the main results in the specific fields? and (3) where are gaps for further research? Publications have focused substantially on the topics of the market, users, data and technology, and the impact on the transportation system. The review shows that, regarding the concept, multi-level approaches have been established for the organization and cooperation of the actors involved, showing different levels of integration of public and private actors on a regional and supra-regional level. Various groups have already been identified as potential users, but the relatively low willingness to pay and the requirements regarding the individualization of mobility services pose problems that have not yet been solved. There is also a constant and unsolved challenge regarding the sensitive motion data that must be stored and processed. Significant research is still needed, including assessments of the impact of MaaS and what effects the service will have on the established use of transport modes, as well as how packages need to be designed and priced in order to optimally reach users.

**Keywords:** mobility as a service; new mobility services; willingness to pay; public transport; mobility user needs; transport ecosystem





## 1. Introduction

The transport sector is a major contributor to greenhouse gas emissions and is therefore facing significant transformations. In addition to more sustainable powertrain technologies for vehicles, mobility as a whole must be rethought and adapted to satisfy its demands without negative outcomes. The large number of new mobility services is an expression of this innovative mood in the mobility market and this has ultimately led to a diversified market landscape, resulting in the development of mobility as a service (MaaS) in 2012 to provide easier access to several services by combining them on one platform.

Due to its novelty, the field of research about MaaS is relatively young. Consequently, individual branches of research are emerging slowly, as publications on specific topics have only started to accumulate in measurable numbers since 2018. At first, publications focused on the question of which features could be used to characterize MaaS and where the service could be differentiated from previous offerings. Meanwhile, focal points in different areas of research became apparent. The aim of this review was to clarify this emerging structure and to assign publications thematically. On the basis of this work, it will thus be possible to draw on existing knowledge in further work.

Expanding upon existing literature reviews [1,2], in this paper we propose an initial classification of the existing literature. Although Arias-Molinares and Garcia-Palomares [1] focused on the definition of MaaS and its components, Utriainen and Pöllänen [2] included publications up to June 2018, when the MaaS literature began to evolve. Based on this foundational and structuring work, this review extends the included literature into the year 2021 and expands on the authors' previous work. The large number of additional papers

included also broadens the range of topics, which are reflected in the structure presented in this paper. This involved examining 127 publications and categorizing them on the basis of their thematic focus. The full range of available publications was included in order to identify currents and research priorities. This distinguishes this review from previously published ones, each of which has placed an emphasis on a specific topic in their analysis.

Arias-Molinares and Garcia-Palomares [1] provided a holistic view of MaaS, reviewing 57 scientific publications [1]. Additionally, a review by Utriainen and Pöllänen [2] analyzed the role of different modes of transport in MaaS and the findings obtained from the first MaaS pilot studies [2]. Both reviews focus on a specific field of research and are therefore of great interest for researchers in these specific areaa. Based on these two papers, in this review we expand the included body of literature and also provide an additional overview of other topics and relevant papers.

### 1.1. Characteristics of MaaS

MaaS plays an essential role in discussions about the future of mobility. Due to the complexity of this field of research, MaaS can be expressed in terms of a currently emerging concept of future mobility, a vision for the future,a new technology, a new user behavior, or a holistic new transport solution [3]. Still, a universal definition of MaaS has not yet been established.

One of the first commercial providers was the company MaaS Global, under the supervision of Hietanen [4], who defined MaaS as a distribution model for mobility that addresses users via a central platform [4]. According to Holmberg et al. [5], MaaS is characterized by a wide range of offered transport services [5]. These vary from peer-to-peer services such as carpooling, to services that optimize the cooperation between public transport and private cars. Smith et al. [6] expanded this view, seeing MaaS as an integrative concept that bundles different transport services into one uniform and seamless service offering [6]. This results in tailor-made mobility solutions, aligned with the individual travel needs of the end user.

Kamargianni et al. [7] continued to narrow the scope and specify that MaaS refers to the offer of bundled services [7]. In this case, the planning, booking, and payment of different forms of mobility through one platform, as well as the availability of mobility packages, were named as important criteria. Other publications have emphasized the focus on individualization and user-orientation. According to Jittrapirom et al. [3], MaaS refers to the pursuit of a transport solution, which meets individual customer needs and offers customized travel connections [3]. This is realized through a user-centered approach with flexible combinations of travel connections and mobility packages, i.e., defined scopes of use of certain means of transport are offered [3,8]. Concurrently, a wider choice of mobility offers enables the customer to be more selective and to have higher demands regarding their desired travel itinerary. In addition to providing a seamless user experience when using various transport services, MaaS serves as a basis for urban efficiency [9]. For this purpose, mobility services, especially means of transportation, are organized according to their time and space efficiency. This is realized via a central mobility platform, which digitally assists and controls travel planning, ticket purchases, and access to transport services, as well as the execution of the journey.

Hensher [10] summarized the demands on MaaS in the form of "three Bs"—bundle, budget, and broker [10]. In contrast to the above-mentioned characteristics such as user-orientation and multi-modality, Hensher [10] considered the innovation of MaaS to be the networking aspect and emphasized the innovations on the side of the provider. A key element of MaaS is the combination of different means of transport with defined conditions of use, which are acquired as part of a package or bundle. The packages are linked to services such as the flexible selection of starting locations and starting times and are offered to the customer in a personalized way, e.g., depending on age, location, or passenger volume [10,11]. Furthermore, as part of the consideration of the budget, it is necessary to balance the preferences of customers and the services offered, relating to a concept

described as willingness to pay. The business model of MaaS can be essentially described using the concept of a broker, as MaaS acts as an intermediary for multi-modal mobility services. In an MaaS ecosystem, the MaaS operator is located between the MaaS users and the transport operators, connecting supply and demand through the provided platform.

Supplementarily, the integration of demand-responsive transportation services in addition to public transport can be seen as critical for the success of MaaS. With the aim of obtaining greater accessibility of existing transport services, demand-responsive services could provide a solution for the first/last-mile problem in the context of the MaaS platform [12,13].

Merging these presented definitions led to the working definition of MaaS used in this paper. By means of MaaS, different types of transport services are integrated into a single mobility service. This service is accessible on demand via a single platform [14]. The platform provides booking, payment, and the provision of access (e.g., tickets) to the end customer via one digital interface. Consequently, the customer receives only one ticket and one invoice for the use of several transport services, as similarly stated by Smith et al. [15].

*1.2. Delimitation of the Term*

MaaS is often associated with the term sharing economy, referring to the shared usage of resources [16,17]. For Frenken and Schor [18], the interaction between private individuals (consumer-to-consumer, C2C) is a defining feature of the sharing economy, whereas Stephany [19] also included business relationships between private individuals and companies (business-to-consumer, B2C) [18,19]. Although private business relationships (C2C) such as ride-hailing are an integral part of many MaaS schemes, others such as car-pooling are not yet part of it [3,7]. In its current development stage, the focus of Maas is plainly on B2C offerings, which can be supplemented with selected C2C services.

Public transport has a central role in all MaaS concepts, which is why integrated public transport (IPT) is often mentioned in the context of MaaS [5]. This refers to an improved coordination and use of different forms of mobility in connection with public transport and is therefore a supporting element for a fully comprehensive MaaS concept [5]. IPT combines several means of transport and thus fulfills an important precondition for MaaS, depending on the public transport situation. Since a complete MaaS scheme allows for any combination of means of transport, the integrated public transport system is only one instrument within a comprehensive MaaS offering [4]. Referring to the classification of mobility services based on their level of integration developed by Sochor et al. [20], IPT forms a basis for the further development of MaaS [20].

Based on this definition and narrowing of the topic, these research questions are explored in the following sections:

- Which topics are discussed in the existing MaaS literature?
- What are the main results in the specific fields?
- Where are the gaps for further research?

The remaining paper is structured as follows. Section 2 presents the methodological approach used in this review. In Section 3, the findings of literature research are displayed and described. Sections 4–7 are dedicated to the different thematic categories and present the relevant findings. Sections 8 and 9 provide an outlook for further research and summarize the main results.

## 2. Methodology

The theoretical approach used for the literature analysis in this work follows the procedure used by Vom Brocke et al. [21], which has already been used in systematic literature analysis on topics of urban mobility and which has proven its fitness [21,22].

The aim of this paper was to describe the key subjects and findings of previous MaaS studies. By identifying and structuring the central research fields, a starting point can be provided for further research, and gaps can thus be identified and specifically addressed. The aim was to achieve a representative overview of MaaS and its developing field of

research; therefore, a neutral perspective was chosen for the presentation of research and no position was taken. The purpose of the analysis was to summarize and synthesize; hence, a conceptual structure is used for the organization of the literature analysis. In this conceptual structure, sources with similar thematic focuses are considered collectively.

Since the term mobility as a service was only established as the dominant term used to refer to combined mobility services as publications progressed, the search term "transport as a service", as well as a combination of generic terms, were included in the database search in Scopus and Science Direct. An overview with the generic terms is presented in Table 1. In accordance with the block-building method of Guba [23] a term matrix with different search terms which are used in the scientific literature as synonyms for MaaS, was set up [23]. In addition to the searches for "Mobility as a Service" and "Transport as a Service", all combinations of the three blocks in Table 1 were queried. Given the number of items in each block ($10 \times 1 \times 3$) and under the restriction that only combinations of all three blocks were chosen, another 30 phrases were formed and used in this study. The selection of search terms was based on a keyword search of the relevant literature. The central element of all links was the word mobility, which was therefore included in every search term.

**Table 1.** Generic terms of the literature analysis.

| Block 1 | Block 2 | Block 3 |
|---------|---------|---------|
| new | | |
| novel | | |
| innovative | | |
| disruptive | | service |
| integrated | mobility | concept |
| connected | | solution |
| combined | | |
| smart | | |
| multimodal | | |
| intermodal | | |

As a time restriction, only papers that were published up to 30 June 2021 were included. MaaS is a young field of research and thus a comprehensive review of its current status, as well as the development of the field of research so far, is presented here. Furthermore, only literature written in German or English was included. Duplicates that occurred when merging the results from the search in the databases were removed.

Conference papers were included as part of the representative overview, but were limited to proceedings to ensure that the latest approaches and findings were included [24]. Secondary literature was included in order to expand the scope of literature considered and to demonstrate a representative state of research. The reference lists of the relevant literature were used to include frequently cited articles. As described above, secondary literature was used to search for relevant keywords on the one hand and relevant literature on the other.

As mentioned in Section 1.1, the scope was limited to the overall concept of MaaS. Articles focusing solely on specific mobility services without the context of being part of MaaS were excluded; thus, the aspect of the combination of services was the core criterion of the included literature. This included, in particular, literature that does not deal with mobility alone, but with complementary issues such as infrastructure, data security, etc., which must inevitably be considered when implementing MaaS.

For the evaluation of the literature, we followed a fixed scheme. Titles and abstracts were first checked to see if they were related to mobility and the previously described containment criteria. This was followed by an analysis of the content of all studies classified as relevant. A pre-sorting of the contents was carried out on the basis of the examined topics, which was expanded and specified in a further iterative process. The categories

formed in this way provided the framework for the content evaluation and sorting of the articles.

The findings are presented within the framework of this article in due brevity in individually thematic chapters and are reduced to central statements. In conjunction with the overview provided in Section 3, this provides a starting point for the further analysis of specific questions in the individual categories.

## 3. Overview of MaaS Literature

### 3.1. Descriptive Data of the Results

The search for all 32 search combinations presented in Section 2 resulted in 1615 hits as shown in Figure 1. This included 501 duplicates. Subsequently, all findings were checked for the relevance of their contentbased on the title and abstract and a further 776 findings were excluded. The remaining 338 publications were intensively checked for their relevance to MaaS. A further 211 articles were then excluded because they did not focus on the topic of MaaS, which ultimately left 127 findings to be included in the classification and this review.

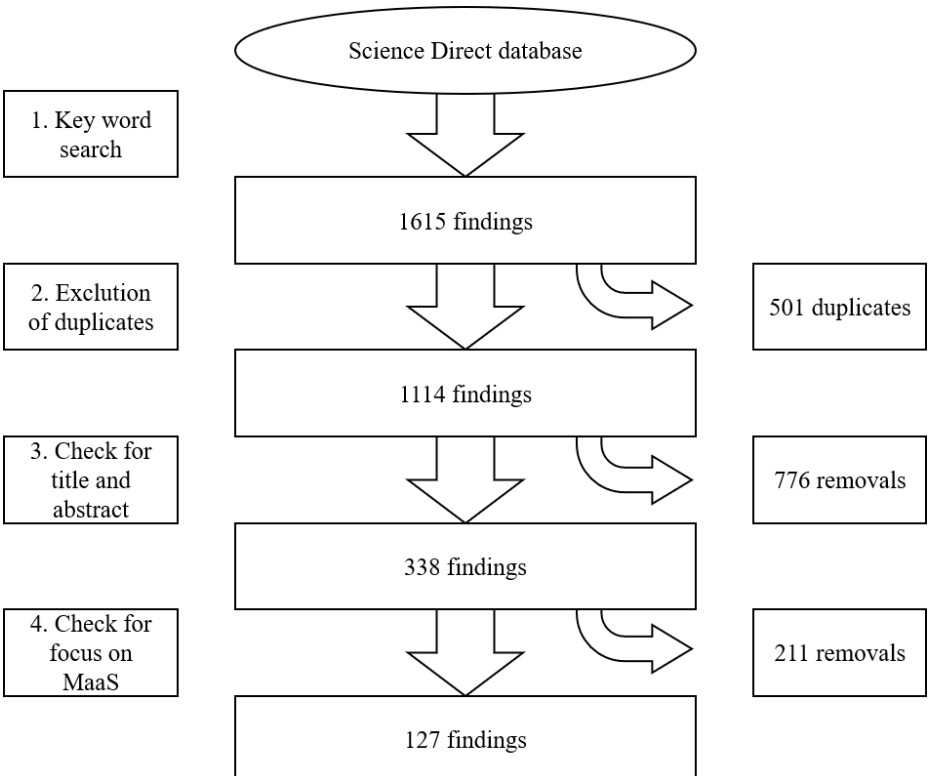

**Figure 1.** Flow chart of included papers.

All results are listed in Table 2 in alphabetical order, referring to the corresponding author. A bottleneck for the reduction of articles proved to be the condition that the publication had to be related to the definitions of MaaS, i.e., a combination of different mobility services using a central platform including uniform billing. Among others, this led to the exclusion of a large number of publications on the topic of smart cards, which generally aimed to simplify the use of public transportation, but not so much the use of various mobility services.

As shown in Figure 2a, most of the publications were conducted in Europe. The country code (CC) of the corresponding author's research location is given in column three in Table 2. The Netherlands, Germany, Sweden, the United Kingdom, and Finland in Europe, as well as Australia, were identified as the centers of international research on MaaS. In total, almost 80% of the publications were created in Europe, which thus occupies the leading position in the research landscape on MaaS. This development was essentially

driven by the origins of MaaS research in Sweden and Finland, which have also been leaders in the implementation and field trials. Due to the wide development of public transport and the multitude of mobility providers that have emerged in the past years, many cities and municipalities in Europe have shown great interest in the dissemination of MaaS. The spiritual father or catalyst of the topic is generally considered to be Hietanen [4], who first launched a commercialized implementation of the concept in Helsinki and later in other European cities with his company, MaaS Global. In recent years, however, the increasing globalization of the research landscape on MaaS has already been observed. Particularly in Asia, alternatives to reduce traffic are increasingly being investigated and tested in response to pressing questions regarding how to solve urban mobility problems.

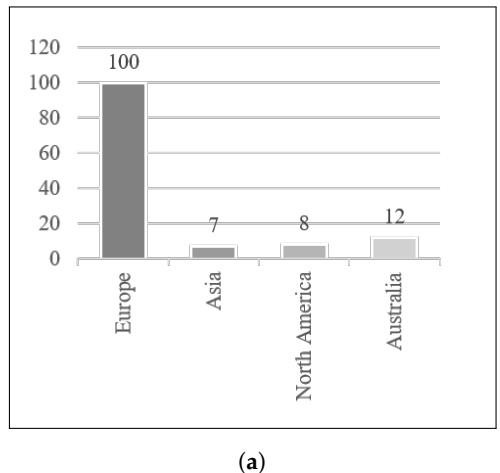

(a)

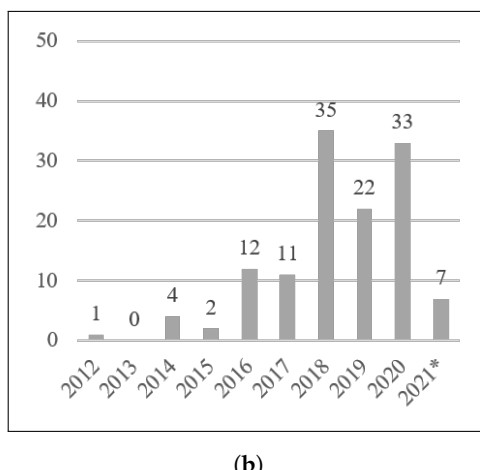

(b)

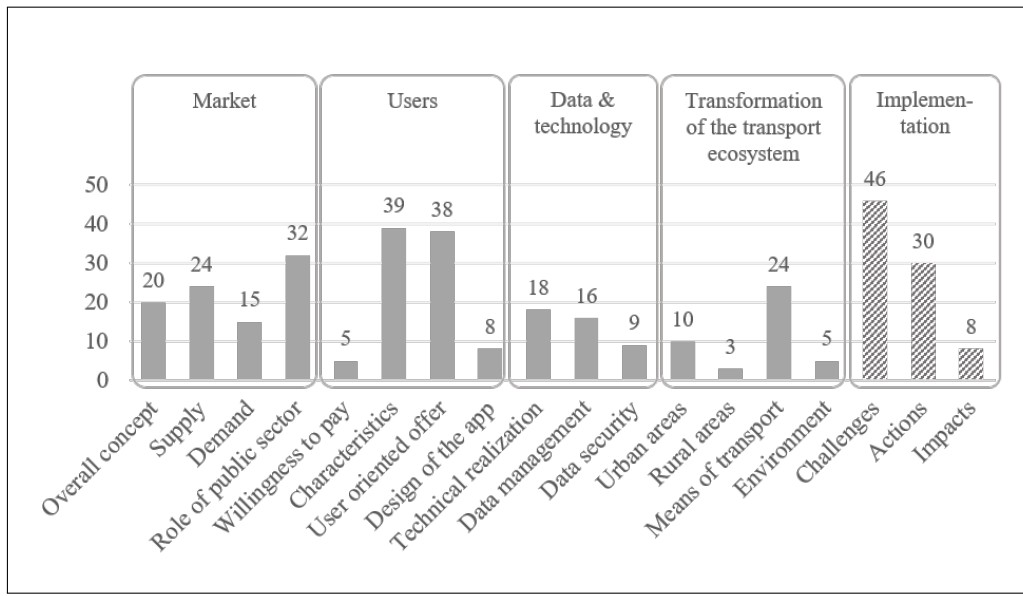

(c)

**Figure 2.** Characteristics of the included literature. (**a**) Origin of publications. (**b**) Number of publications in each year. (* only until June 30th) (**c**) Topics of the publications.

Figure 2b shows the development of publication figures in recent years, which increased significantly for the first time in 2016 and again in 2018. An additional increase was seen in 2020, when a total of 33 papers were published. Up to June 2021, another seven papers were published. This development indicates the increased interest in the topic and is reflected in a broader and more diverse research landscape. In parallel, the excluded findings also showed a significant increase in publications after 2016, demonstrating the increased research interest in issues related to mobility and alternative mobility services.

**Table 2.** Overview of the MaaS literature.

| Author | Year | CC | DEF | Market | | | | | Users | | | Data | | | Transport Ecosystem | | | | Implementation | | |
|---|---|---|---|---|---|---|---|---|---|---|---|---|---|---|---|---|---|---|---|---|---|
| | | | | OC | S | D | RPT | WTP | C | DA | US | TR | DM | DS | UA | RA | MT | E | C | A | I |
| Aditjandra [25] | 2019 | UK | | | | | | | | ● | | | | | | | | | | | |
| Alonso-Gonzalez et al. [26] | 2020 | NL | | | | | ○ | | ○ | ● | | | | | | | | | | | ○ | |
| Alyavina et al. [27] | 2020 | UK | | | | | | | ● | ● | | | | | | | ● | ○ | | | |
| Ambrosino et al. [28] | 2016 | IT | | | | | | | | | | ● | ○ | | | | | | | | |
| Arias-Molinares and Garcia-Palomares [29] | 2020 | ES | | ○ | ○ | | ○ | | | | | | | | ○ | | | | ● | | |
| Arias-Molinares and Garcia-Palomares [1] | 2020 | ES | ● | | ● | ○ | ● | | ○ | | ○ | | | | | | | | ● | ○ | |
| Audouin and Finger [30] | 2018 | FI | | | | | ● | | | | | | | | | | | | ○ | ○ | |
| Barreto et al. [31] | 2018 | PT | | | | | | | | | | | | | | | ● | | ● | ○ | |
| Barreto et al. [32] | 2018 | PT | | | ○ | | | | | | | ○ | ○ | ○ | | | | | ○ | | |
| Baumann and Püschner [33] | 2017 | DE | | | | | | | | | | ● | ○ | | | | | | | | |
| Becker et al. [34] | 2019 | CH | | | ○ | | | | | | | | | | | | | ● | | | ○ |
| Bellotti et al. [35] | 2016 | IT | | | | | | | | | | | ○ | | | | | | | | |
| Beutel et al. [36] | 2014 | DE | | | | | | | | ● | | | | | | | | | | | |
| Beutel et al. [37] | 2018 | DE | | | | | | | | | | ● | | | | | | | | | |
| Beutel et al. [38] | 2019 | DE | | | | | ○ | | | ○ | | ○ | | | | | | | | | |
| Brendel and Mandrella [24] | 2016 | DE | | | | | | | | | | ● | | | | | | | | | |
| Caiati et al. [13] | 2020 | NL | | ○ | | ○ | ● | | | ● | ● | ● | | | | | | | | | ● | |
| Calderón and Miller [12] | 2019 | CA | | ○ | ○ | ● | | | | | | | | | | | | | ○ | ● | |
| Callegati et al. [39] | 2017 | IT | | | | | | | | | | ● | | | | | | | | | |
| Callegati et al. [40] | 2018 | IT | | | | | | | | | | | | ● | | | | | ● | ● | |
| Casadó et al. [41] | 2020 | UK | | | | | | | ○ | ○ | | | | | | | | | ○ | | |
| Chang et al. [42] | 2019 | TW | | | | ● | ○ | | ○ | | | | ○ | | ○ | | | | | | ○ |
| Cisterna et al. [43] | 2021 | LU | | | | | | | | | | | | | | | ● | | | | |
| Cooper et al. [44] | 2019 | UK | | | | | | | | | | | | | ○ | | ○ | | | | |
| Cottrill [45] | 2020 | UK | | | | | | | | | | | ● | ● | | | | | | | |
| Djavadian and Chow [46] | 2017 | CA | | | | ● | | | | | | | | | | | | | | | |
| Docherty et al. [47] | 2017 | UK | | ● | | | ● | | | | | | | | | | | | | | |
| Durand et al. [16] | 2018 | NL | | | | | | | ○ | ● | | | | | | | | ○ | | ○ | |
| Eckhardt et al. [48] | 2018 | FI | | | | | | | | | | | | | | | | ● | ● | ● | |
| Esztergár-Kiss and Kerényi [49] | 2019 | HU | | | | | | | | ● | | | | | | | | | | ● | ○ | |
| Feneri et al. [50] | 2020 | NL | | | | ● | | | ● | ● | | | | | | | | | | | |
| Fenton et al. [51] | 2020 | SW | | ○ | | | ● | | | | | | | | | | | | | ○ | ○ | |
| Fioreze et al. [52] | 2019 | NL | | | | | ○ | | ○ | ● | | | | | | | | | | | |
| Frenken and Schor [18] | 2017 | NL | | | | | | | | | | ○ | ○ | | | | | | | | |
| Fröhlich et al. [53] | 2018 | AT | | | | | | | ○ | ○ | | | | | | | | ○ | | ○ | |
| Gonzalez-Feliu et al. [54] | 2018 | FR | | | | ● | | | | | | | | | | | | | | | |
| Guidon et al. [55] | 2020 | CH | | | | | | | ● | ○ | | | | | | | | | | | |
| Harrison et al. [56] | 2020 | UK | | ● | | | | | | | ○ | ● | | | | | ○ | | ○ | ○ | |

**Table 2.** *Cont.*

| Author | Year | CC | DEF | Market | | | | | Users | | | | Data | | Transport Ecosystem | | | | Implementation | | |
|---|---|---|---|---|---|---|---|---|---|---|---|---|---|---|---|---|---|---|---|---|---|
| | | | | OC | S | D | RPT | WTP | C | UO | DA | TR | DM | DS | UA | RA | MT | E | C | A | I |
| Hasegawa [57] | 2018 | JP | | | | ● | | | | | | | | | | | | | | | |
| Hawkins and Habib [58] | 2019 | CA | | | | | | | | ● | | | | | | | | | | | |
| He et al. [59] | 2018 | US | | | | | | | | | | | | ● | | | | | ● | ● | |
| Hensher [10] | 2017 | AU | | | | | | | | | | | ○ | | ○ | | ● | ○ | ○ | | |
| Hensher et al. [60] | 2021 | AU | | | | | | | ● | ● | ● | | | | | | ○ | | | | |
| Hesselgren et al. [61] | 2020 | SW | | | | | | | ● | ● | ● | | | | | | ○ | | | | |
| Hietanen [4] | 2014 | FI | | | | | | | ● | | | | | | | | | | | | |
| Hilgert et al. [62] | 2016 | DE | | | | | | | | ● | | | | | | | | | ○ | ○ | |
| Hirashima et al. [63] | 2019 | JP | | | | | | | | | | | ● | | | | | | | | |
| Hirschhorn et al. [64] | 2020 | NL | | ● | ● | | ● | | | | | | | | | | | | | | |
| Ho et al. [11] | 2018 | AU | | | | | | | ○ | ● | | | | | | | ○ | | ● | ● | |
| Ho et al. [65] | 2020 | AU | | | | ● | | ● | ○ | ● | | | | | | | | | | | |
| Hörcher and Graham [66] | 2020 | UK | | | | ○ | | | | | | | | | ● | | ● | | ○ | | |
| Hoerler et al. [67] | 2020 | CH | | | | | | | ● | ● | | | | | | | | | ○ | ○ | |
| Holmberg et al. [5] | 2016 | SW | ● | ● | | ○ | ○ | | ○ | | | | | | | | | | ● | | |
| Jang et al. [68] | 2021 | NL | | | | | | | ○ | ● | | | | | | | | | ● | | |
| Jin and Qiu [69] | 2019 | SG | | | | | ○ | | | | | | ○ | | | | | | ○ | | |
| Jittrapirom et al. [3] | 2017 | NL | | ○ | ○ | | | | | | | | | | | | | | ○ | | |
| Jittrapirom et al. [70] | 2018 | NL | | ● | | | | | | | | | | | | | | | | | |
| Jittrapirom et al. [71] | 2018 | NL | | ○ | | | | | | ● | | | | | | | | | ○ | ○ | |
| Kamargianni et al. [72] | 2015 | UK | | | | | | | | | | | | | | | | | | | |
| Kamargianni et al. [7] | 2016 | UK | | | ● | | | | | | | | | | | | | | | | |
| Kamargianni and Matyas [73] | 2017 | UK | | | | | ● | | | | | | | | | | | | | | |
| Karinsalo and Halunen [74] | 2018 | FI | | | | | | | | | | ● | | | | | | | | | |
| Karlsson et al. [75] | 2016 | SW | | | | | ○ | | ● | | | | | | | | ○ | | ○ | | ● |
| Karlsson et al. [76] | 2020 | SW | | ● | ○ | ○ | ● | | | | | | | | | | | | | | |
| Keller et al. [77] | 2018 | DE | | | | | | | ● | | | | | | | | | | ○ | ○ | |
| Kim et al. [78] | 2021 | SK | | | | | | | ● | | | | | | | | ● | | | | |
| Kortum [79] | 2016 | US | | | | | | | ● | ○ | | | | | ○ | | | | ○ | | |
| Lang and Mohnen [80] | 2019 | DE | | | | | | | | ○ | | | ○ | | | | ○ | | | | |
| Lerner and Van Audenhove [81] | 2012 | DE | | | | ○ | ○ | | | | | | | | | | | | ○ | ○ | |
| Li et al. [82] | 2019 | UK | | ● | | | | | | ○ | | | | | | | | | ● | | |
| Liljamo et al. [83] | 2020 | FI | | | | | | ● | | | | | | | | | | | | | |
| Lopez-Carreiro et al. [84] | 2020 | ES | | | | | | | ● | ● | ● | | | | | | | | | | |
| Lucken et al. [85] | 2019 | US | | | | | ● | | | | | | | | | | | | | | |
| Lyons et al. [86] | 2019 | UK | | | ○ | | | | ● | | | | | | | | ○ | | ○ | | |
| Lyons [87] | 2020 | UK | | | | | | | | | | | | | ● | | ● | | | | |
| Marchetta et al. [88] | 2016 | IT | | | | | | | | ● | | | ○ | ○ | | | | | | | |
| Matyas [89] | 2020 | UK | | | | | | | ● | ● | | | | | | | ○ | | | | ○ |

**Table 2.** *Cont.*

| Author | Year | CC | DEF | Market | | | | | Users | | | | Data | | Transport Ecosystem | | | | Implementation | | |
|---|---|---|---|---|---|---|---|---|---|---|---|---|---|---|---|---|---|---|---|---|---|
| | | | | OC | S | D | RPT | WTP | C | UO | DA | TR | DM | DS | UA | RA | MT | E | C | A | I |
| Matyas and Kamargianni [90] | 2021 | UK | | | | | | | ● | ● | | | | | | | | | | | |
| Matyas and Kamargianni [91] | 2018 | UK | | | ● | | | | | | | | | | | | | | ○ | | |
| Melis et al. [92] | 2018 | IT | | | | | | | | | ○ | ● | | | | | | | | | |
| Merkert et al. [93] | 2020 | AU | | | ● | | | | | | | | ○ | ○ | | | | | | | |
| Meurs et al. [94] | 2020 | NL | | ● | ● | | ● | | | | | | | | | | | | | ● | |
| Miramontes et al. [95] | 2017 | DE | | | | | | | | | | | | | | | ● | | | | |
| Mladenović and Haavisto [96] | 2021 | FI | | ○ | ○ | ○ | ● | | | | | | | | | | | | | | |
| Mukhtar-Landgren and Smith [97] | 2019 | SW | | | | | ● | | | | | | | | | | | | ● | ○ | |
| Mulley et al. [98] | 2020 | AU | | | | ○ | | | ● | | | | | | | | | | | | |
| Mulley et al. [99] | 2018 | AU | | | | | | | ○ | ○ | | | | | | | | | | | |
| Mulley and Kronsell [100] | 2018 | AU | | | | | ● | | | | | | | | | | | | | | |
| Pangbourne et al. [8] | 2018 | UK | | | | | | | ● | ○ | | | ○ | | | | ○ | ○ | ○ | | |
| Pantelidis et al. [101] | 2020 | US | | | | | | | | | | ● | | | | | | | | | |
| Pflügler et al. [102] | 2016 | DE | | | | | | | | | | | ● | | ○ | | | | | | |
| Pickford and Chung [103] | 2019 | HK | | ○ | ● | | | | | | | | | | | | | | | | |
| Polydoropoulou et al. [104] | 2018 | GR | | | ○ | | | | ○ | | | | ○ | ○ | | | | | | ● | |
| Polydoropoulou et al. [105] | 2019 | GR | | ○ | | | ○ | | | | | | | | | | | | | ○ | |
| Reck et al. [106] | 2020 | SW | | | | | | | | ● | | | | | | | | | | | |
| Rizzoli et al. [107] | 2014 | CH | | | | | | | ○ | ○ | | | | | | | | | | | |
| Russ and Tausz [108] | 2015 | AT | | | | | ● | | | | | | | | | | | | | | |
| Schwinger and Krempels [109] | 2019 | DE | | | ● | | | | | | | | | | | | | | ○ | ○ | |
| Smith et al. [110] | 2017 | SW | | | | | ● | | ○ | | | | | | | | | | ○ | ○ | |
| Smith et al. [111] | 2018 | SW | | ○ | | | ● | | | | | | | | | | ● | | | | ○ |
| Smith et al. [6] | 2018 | SW | | | | | ● | | | | | | | | | | | | ○ | ○ | |
| Smith et al. [112] | 2019 | SW | | | | | ● | | | | | | | | | | | | ● | ○ | |
| Smith et al. [113] | 2020 | SW | | ● | ○ | | ○ | | | | | ● | ● | | | | | | | ○ | |
| Smith and Hensher [114] | 2020 | SW | ● | ● | | | ● | | | | | | | | | | | | | | |
| Sochor et al. [20] | 2018 | SW | | | | | ● | | | | | | | | | | | | | | |
| Stopka [115] | 2014 | DE | | | | | | | ○ | ● | | | ○ | | | | | | | | |
| Stopka et al. [116] | 2018 | DE | | | | | ○ | | ○ | ○ | | | | | | | | | | | |
| Storme et al. [117] | 2019 | BE | | | | | | | ○ | | | | | | | | ○ | | | | ○ |
| Strömberg et al. [118] | 2018 | SW | | | | | | | ● | | | | | | | | ○ | | ○ | ○ | |
| Surakka et al. [119] | 2018 | FI | | | | | ● | | | | | | | | | | | | ○ | ○ | |
| Tsouros et al. [120] | 2021 | GR | | | | | | | ● | ● | | | | | | | | | | | |
| Utriainen and Pöllänen [2] | 2018 | FI | | | | | | | ○ | | | | | | | | ● | | | | ○ |
| Veeneman et al. [121] | 2018 | NL | | | | | | | | | ● | ● | ○ | | | | | | | | |
| Vij et al. [122] | 2018 | AU | | | | | | | ○ | ● | | | | | | | | | | | |

**Table 2.** *Cont.*

| Author | Year | CC | DEF | Market | | | | | Users | | | | Data | | Transport Ecosystem | | | | Implementation | | |
|---|---|---|---|---|---|---|---|---|---|---|---|---|---|---|---|---|---|---|---|---|---|
| | | | | OC | S | D | RPT | WTP | C | UO | DA | TR | DM | DS | UA | RA | MT | E | C | A | I |
| Wienken and Krömker [123] | 2018 | DE | | | | | | | | ● | | | | | | | | | | | |
| Wittstock and Teuteberg [124] | 2019 | DE | | ○ | ○ | | | | | | | | | | | | | | ● | ○ | ○ |
| Wong et al. [125] | 2020 | AU | | | ● | | | | | | | | | | | | | | | | |
| Wong et al. [9] | 2018 | AU | ● | | | | ○ | | | ○ | | | | | | | | | | | |
| Wright et al. [126] | 2020 | UK | | | | | | | | | | | | | ● | | ● | | | | |
| Wu et al. [127] | 2018 | AU | | | | | | | | | | | ● | | | | | | ● | ● | |
| Yuan et al. [128] | 2016 | US | | | | | | | | | | | | ● | | | | | | | |
| Załoga and Wojan [129] | 2017 | PO | | | | | ● | | | | | | | | | | | | ○ | ○ | |
| Zhang et al. [130] | 2018 | JP | | | | | | | | | | | | | | | ● | | | | ○ |
| Zhao et al. [131] | 2020 | SW | | | | | | | ● | ○ | | | | | | | | | | | |
| Zijlstra et al. [132] | 2020 | NL | | | | | | | ● | | | | | | | | | | | | |

Notes: Abbreviations in header are as follows: CC—country code; DEF—definition; OC—overall concept; S—supply; D—demand; RPT—role of public transport; WTP—willingness to pay; C—characteristics; UO—user-oriented offers; DA—design of the app; TR—technical realization; DM—data management; DS—data security; UA—urban areas; RA—rural areas; MT—means of transport; E—environment; C—challenges; A—actions; I—impacts; ●—focus of the paper; ○—partly discussed in the paper.

### 3.2. Topics Discussed in the MaaS Literature

To categorize the findings, four major fields of research were identified, namely, the market, users, data and technology, as well as the transformation of the transport ecosystem. In addition to these four thematic pillars, recurring implementation issues emerged in the articles studied, which can be considered as levels of implementation. The identification of the type and scope of the implementation has been carried out in conjunction with the discussion of the content of the articles, which is why a separate chapter was not necessary for this purpose. Each category contains multiple subcategories, which are evaluated more intensively in the respective chapters. A strict separation of the topics from one another was not always possible and was also not sought. Rather, the overview provided here is intended to express the priorities of the individual works in order to take them into account in the context of further analysis.

As shown in Figure 2c, not all topics were discussed as frequently as others. Market design issues were the most frequently addressed, with the overall concept being examined in 20 publications, the design of the supply and demand sides in 24 and 15, respectively, and the role of the public sector in 32 publications. Above all, the relationship between public and private actors was discussed, which is reviewed in greater detail in Section 4.

The users of MaaS and the question how they can be reached in a targeted manner formed the second-largest group of topics in the publications examined. In 39 articles users of MaaS were analyzed based on surveys or their usage behavior, and 38 articles presented user-specific offers with which certain user groups could be reached. The design of apps and willingness to pay were examined much less frequently, in eight and five publications.

Topics in the data and technology category were taken up a total of 43 times, with 18 publications dealing with questions relating to the technical implementation of an MaaS system and 16 publications analyzing data management. Data security issues were discussed in nine publications.

The fourth thematic category was topics related to the transformation of the existing transport system. Ten and three publications discussed the changes in urban and rural areas, respectively. Twenty-four publications dealt with the question of which modes of transport are needed for MaaS and another five articles discussed the effects on the environment.

At the level of the implementation of MaaS, 46 articles addressed challenges in the respective area associated with MaaS. Another 30 identified specific actions that must be taken to implement MaaS. Eight articles again analyzed which specific effects could be expected with the implementation of MaaS.

Table 2 includes all 127 publications and the relevant information about the author, year of publication, and the country code of the corresponding research institution. The additional columns are the result of the structuring of content. For reasons of visualization, the header of the table contains only the main categories and uses deprivations for subcategories.In order to visualize the depth of the discussion of a topic in the context of an article, two types of symbols were used. Fully completed Harvey balls symbolize a strong focus on the topic in the context of the publication. Empty Harvey balls, on the other hand, symbolize a discussion of the topic, but this is not the focus of the article. To ensure a correct understanding of the table, the following abbreviations are used (these are provided again under the table).

- CC—country code of corresponding author;
- Market: OC—Overall Concept, S—Supply, D—Demand, RPT—Role of Public Transport;
- Users: WTP—Willingness to pay, C—Characteristics of users, UO—User-oriented Offers, DA—Design of app;
- Data and technology: TR—Technical Realization, DM—Data Management, DS—Data Security;
- Transformation of the transport ecosystem: UA—Urban Areas, RA—Rural Areas, MT—Modes of transport, E—Environment; and
- Implementation: C—Challenges, A—Actions, I—Impacts.

## 4. Market

*4.1. Overall Concept*

Although Li et al. [82] emphasize in their work that MaaS concepts certainly follow a uniform characteristic and are not constructed according to a single business model [82], Karlsson et al. [76] provide an in-depth analysis of the overall MaaS concept and develop an analytical framework of the included players and their impact on MaaS [76]. For further development and integration of MaaS, action is needed on a macro-, meso-, and micro-level, firstly by setting a matching policy to support and foster MaaS [76]. Legislators at a national level, as well as regional authorities and private players, have a duty to develop and set a common regulatory framework, in addition to the one practiced at the beginning of the development of MaaS with players at the micro-level [108].

In addition to the importance of structured and reliable legislation, three basic types of organization of MaaS concepts depending on the administration are discussed, namely, a market-driven, publicly controlled, or combined private-public approach, whereas private and public partnerships are seen as the ones with the highest chances of success [6,56,94]. Smith et al. [113] support that view and stress that without public mobility providers MaaS cannot be realized [113]. The authors propose the idea of MaaS integrators to facilitate the development and implementation of MaaS. So-called intermediary MaaS integrators (IMIs) collect offerings from transport service providers (TSPs) and distribute them to the operating MaaS providers. By focusing on the technical aspects and the handling of data, the use of IMIs reduces some of the main challenges involved in collaboration between transport and MaaS providers. Still, the use of IMIs is not limited to the technical aspect; rather, they supports the overall development of MaaS and reduce the presence of barriers in the development process. Polydoropoulou et al. [105] support this role of public transport providers, but also highlight concerns about the ability of transport authorities to play such a role due to structural and financial constraints [105].

A solution to address these concerns was provided by Pickford and Chung [103], who proposed a path to reduce the challenges involved in the development of MaaS by means of the concept of MaaS Lite [103]. Instead of developing a one-size-fits-all solution, the authors focus on first developing a smaller variety of services and transport operators under the umbrella of MaaS. Step by step, and with the increasing acceptance of users, more services can be integrated into the service. More participating passengers means an added value for (potential) providers and this leads to more providers in the market, which expands the available options for customers. The same focus was promoted by Jittrapirom et al. [3], who identified the merging of supply and demand onto one unified platform and a critical mass of market participants as being critical for the concept [3].

*4.2. Matching Supply and Demand*

To achieve as many users as possible, supply should be derived from the patterns of daily activities of potential users [3,109]. However, the new opportunities offered by MaaS are countered by a high proportion of fixed assets in the transport industry, as well as strong barriers, especially the unwillingness of transport operators and MaaS operators to cooperate with each other [6,86,124]. Transport ecosystems are currently highly fragmented and characterized by protectionism and risk aversion among transport service providers. This leads to an unwillingness to allow third parties to resell tickets, a lack of available data, and general uncertainty about emerging MaaS business models. In contrast, the key to overcoming protectionism and risk aversion among different transport service providers may be the creation and iterative adaptation of a general vision, including deregulation and regulation for MaaS.

To overcome these barriers and support the complex decision-making process involved when purchasing MaaS projects, Matyas and Kamargianni [91] set up a survey design for local transport providers [91,133]. They emphasized that despite the potential for market entry, the local transport provider should first be examined in detail. This includes the collection of information on organization, pricing and offer design, and possible

cooperation, as well as existing combined offers from several operators. Next, the role of the public transport provider in the concept of MaaS needs to be clarified. Various forms of participation are possible, ranging from the mere financing of the service provider to the provision of intangible know-how or tangible assets such as vehicles and depots [134]. Section 4.3 provides a deeper look into this subject.

For private mobility providers, the motivation to participate in MaaS is characterized by a growing sales potential and market share [104]. This is supplemented by access to high-quality demand-related data. In this context, public transport providers must examine to what extent their own goals coincide with the interests of private providers. To achieve a high level of collaboration, a mutual information exchange is necessary between different interest groups, such as transport companies, customers and authorities [54].

To evaluate different MaaS services with different features and offerings, Kamargianni et al. [7] developed a reference index [7]. The degree of integration describes the scope of the MaaS offer and is based on three main elements that enable seamless intermodal journeys for users, namely, ticket integration, the integration of information and computer technology (ICT), and the integration of mobility packages. Sochor et al. [20] and Lyons et al. [86] adopted the level-of-integration model and proposed a topology that enables the positioning of mobility services in the MaaS context, whereby the decisive factor is also the scope of services and thus the functions made available to the user. This scope of services is broken down into different levels of integration (e.g., according to the type of mobility service), resulting in five levels: 0—no integration, 1—integration of information, 2—integration of booking and payment, 3—integration of service offers such as contracts, and 4—integration of social objectives.

In addition to the design of the systems, the literature also discusses systems for controlling an ongoing MaaS system and how these can permanently ensure a balance of supply and demand. Flexible prices are conceivable as possible system variables for control, which can ensure the high utilization of various modes of transport according to the current demand using ICT [57,81]. However, systems developed for optimizing these systems should always include both sides of the market in the decisions, since two-sided markets are characterized by the mutual influence of buyers and sellers. Not only prices, but also the scarcity or expansion of supply, can be used as control instruments in this case [46].

### 4.3. Role of the Public Sector

The public sector is of great importance for the development of MaaS [64,96]. The business model of MaaS challenges established rules of regulation, since both public and private companies are involved. The changed conditions and players in the mobility market make it necessary to rethink various traditional regulations in the mobility market [47,114]. The majority of the actors involved would like the local and regional administration to play an active role in the development and shaping of legal frameworks and in the provision of resources, since their support is essential for the development of MaaS [51,97]. In the literature, however, there are different views on the nature and extent of the involvement of local authorities. The scope ranges from soft factors such as promotion or supporting tasks in the development of MaaS projects [110] to financing and concluding binding contracts with actors [97].

A consensus that emerged in the publications was the formulation and specification of a goal or a roadmap for the development of MaaS by local authorities [30,79,97,110]. Only by creating a uniform target image of MaaS with a long-term orientation can the necessary framework conditions be created so that public transport and private actors can join forces in collaborative projects [96,129]. The distribution of roles and responsibilities between actors in this emerging ecosystem is a weakly framed but significant issue. Thus, it is seen as a major challenge to motivate public and private actors to agree on new common contractual conditions [110].

Pangbourne et al. [8] examined the impact of MaaS with regard to the role of the public sector and also viewed the allocation of liabilities in a complex network of stakeholders as an important step toward establishing the MaaS concept in society [8]. Thus, consumer protection should be defined and market rules, such as minimum standards for services, should be developed.

However, interviews among MaaS stakeholders have shown that transport authorities are not perceived as promoters of collaborative innovation at the organizational level and that the organizational culture of transport authorities encourages inertia and prevents experimental approaches. The quasi-monopolistic position of the local transport operators thus poses long-term risks to the concept as a whole [104]. At the internal level it becomes clear that innovation efforts within transport authorities are hampered by a lack of human capital and by the low prioritization of MaaS development [112]. As a consequence, innovation-friendly environments should be allowed to support pilot projects and research [6]. In such a working environment, all relevant stakeholders must be integrated into the development process and, in particular, political actors must be instructed accordingly [108].

### 4.3.1. Cooperation between Market Players

In addition to the MaaS integrator, a large number of actors are involved in the design of an MaaS system: transport companies, data providers, technical service and IT providers, ICT infrastructure, insurance companies, regulatory authorities, universities and research institutions, and customers [73]. Coordination and alignment among actors is crucial for the success of MaaS implementation and this is one of the biggest challenges [125]. Therefore, for the transition to an MaaS-based transport system, a multi-stakeholder approach is needed, in which a network of players acts together and induces the necessary institutional changes [6]. For this purpose, a basic understanding of existing institutional structures (legislation, politics, culture) and practices (experiments, cooperation) is essential to derive relevant actors and recommendations for action.

Apart from the frequently mentioned promotion of cooperative relations between the public sector, i.e., public transport companies and municipalities, and private companies, the relations between the private companies themselves should be strengthened [105]. Only if the providers of mobility services are willing to cooperate can the potential advantages of MaaS for customers and cities be exploited.

Smith et al. [112], as well as Załoga and Wojan [129], highlight that the public sector should ensure the active participation of potential end users in the innovation process, since their acceptance and ideas ultimately determine the success of the innovation. The involvement of customers is an important component of MaaS development, meaning that the cooperation between private and public actors can be extended to a public–private partnership [108].

Long-term commitments by public authorities—beyond pilot projects—could result in increased motivation and attractiveness for private investors [6]. Therefore, it is beneficial to encourage public stakeholders to focus on the long-term effects of disruptive innovations when funding MaaS projects. However, they should also allow more leeway for short-term objectives [112].

### 4.3.2. Competing Objectives of Market Players

Due to their divergent motivations, public and private mobility providers have contradictory objectives. Although private operated mobility services generally aim to maximize their revenue, public transport operators strive for social benefits such as easy, accessible, and affordable mobility, as well as improvements in the quality of life for residents, e.g., through less congestion or less environmental pollution [6,116,119].

In addition, laws and regulations make it difficult for transport authorities to establish long-term partnerships with private actors that go beyond their actual task of providing public transport [6]. Too much regulation would also affect the innovation performance of

private actors, leading to unattractive MaaS offerings. However, too little regulation could lead to a form of MaaS that does not serve the public interest. Appropriate regulation, as well as incentives, play key roles in relation to the public sector, facilitating the development of workable and sustainable MaaS concepts.

Addressing this conflict of interest, Mulley and Kronsell [100] conducted workshops with scientists, operators, and decision-makers to discuss how new mobility services can be integrated into the market. As a result of the workshops, it became clear that transport authorities have to create new structures in an MaaS environment and revise existing onesm such as budget planning, especially if it is assumed that MaaS will not make public transport more profitable [100]. Public-private partnerships are able to improve the efficiency and quality of mobility services, but should be considered carefully, since they compete with existing public transit services [85].

Nevertheless, private partners and MaaS itself profit from a close connection to public transport providers, since existing public transport brands are perceived as trustworthy. Moreover, a lot of time and financial resources would be required for people to gain awareness of a novel brand. Nevertheless, it has been argued that a combination of the introduction of a new brand and a well-known existing brand would represent the optimal offer for the customer [110].

## 5. MaaS Users

### 5.1. Characteristics of Customers

Personal attitudes and value orientations, as well as different socioeconomic resources, lead to the pluralization of customer requirements and needs in transportation, and eventually to MaaS [53]. Consequently, the demand for individualized services is constantly increasing [99]. Travelers wish to have a choice between different modes of transport and despite the demand for more freedom of choice, an expectation of high autonomy and temporal and spatial flexibility, as well as reliability, remains [16,80,104]. Therefore, MaaS operators are confronted with a multitude of expectations.

However, it should be noted that if users are satisfied with their existing mobility situation, there is no motivation to reconsider their behavior, especially for regular users of private cars [65,78] or frequent users of public transport [11,86,132]. The simple supply of MaaS does not automatically generate user demand. Potential customers must be shown the improvements in the framework conditions, for example, in financial and environmental terms, in order to trigger the consideration of the MaaS offer [86].

As one of the few fully functional and scientifically covered MaaS projects, UbiGo in Sweden provided insights into the users of MaaS and their expectations. Although the participants systematically overestimated their transportation needs at the beginning of the project, 64% of them changed their use of mobility during the project towards more sustainable or shared modes [75]. Strömberg et al. [118] referred to the same project, stating that users had different expectations regarding the mobility service offered [118]. Therefore, for the optimal design of MaaS, user evaluations must be carried out to better understand the relationships between service offerings, customer satisfaction, and environmental impacts in relation to different design types [118].

A large number of studies were also devoted to the characterization of MaaS users. An important influencing factor was age, as mainly younger and middle-aged groups used MaaS [16,67,72,90,122]. Furthermore, the family situation was found to play a major role, as families with children used MaaS significantly less frequently and used cars more frequently [11,60,90]. In addition, higher education, regular use of car sharing, a regular income, and an ecological attitude, as well as the desire to reduce car use, influenced the intention to use MaaS [26,67,116]. Furthermore, if experience had already been gained with multimodal platforms, the likelihood that users would use MaaS increased [79] and it decreased in the absence of experience [52].

The role of young adults as a target group was questioned by Jittrapirom et al. [71], since the lower purchasing power of this group means that MaaS must be checked in terms

of pricing and general affordability [71]. In terms of future users of MaaS, Casadó et al. [41] highlighted the influence of parental behavior on children and how they chose modes of transport [41].

### 5.2. User-Oriented Services

As stated before, MaaS operators are confronted with a multitude of expectations regarding MaaS. However, individual user characteristics can be used to bundle similar expectations and thus target groups of users. In addition, users have clear requirements regarding individual forms of transport. Individual modes can be essential for users, i.e., absolutely necessary, considerable, or even eliminating if they are consistently rejected [89]. Bundles of modes of transport are called MaaS packages or plans.

The idea behind mobility plans is that several combined services generate greater added value for customers than individual separate services. The use of transport in a package can encourage customers to use services they would not normally use [99] and it offers possibilities for providers to encourage the use of specific modes of transport [16,68]. When properly combined, this can lead to an increase in the frequency of sustainable transport usage [72]. Because of the great heterogeneity in the demand for mobility packages, Kamargianni et al. [72] recommend a dialogue with end users in the sense of collaborative customization [72]. Reck et al. [106] provide a framework in their study that enables the comparable creation of MaaS bundles and ensures that even different regions deliver comparable results for research through uniformly created bundles [106]. In order to make the packages as customer-oriented as possible, individual data are required. Information such as age, occupation, and family status, together with data on travel behavior, helps to address the needs of customers precisely and to personalize the composition of mobility packages [32,68,72]. In addition to the combination of different mobility services, additional services must also be included in the creation of the packages, for example, real-time data on the capacity utilization of means of transport [84]. Due to the sensitive nature of the data, the active consent and cooperation of the customers is a basic prerequisite for putting together optimal mobility packages [16].

### 5.3. Willingness to Pay and Payment Models

A comprehensive approach to the determination of user preferences was carried out by Caiati et al. [13]. Although the results of the survey confirmed the role of public transport, the study provides many other interesting insights into the choice of mobility packages. On the one hand, the general tendency towards subscriptions in the transport sector is low. However, packages with flat rates, i.e., unlimited use, or which consist of only two components, can achieve greater acceptance. In addition, the authors include the role of the social environment in the choice of package, which can have a measurable influence. Although the results have to be viewed in relation to the ongoing discussion in the Netherlands, relevant factors for the package composition are already evident in these studies [65,135].

A large discrepancy between stand-alone services and bundles of mobility services was highlighted by Guidon et al. [55]. Bike and e-bike sharing, as well as taxis, were related with a negative willingness to pay when included in a bundle, whereas car sharing, park and ride services, and public transportation were associated with a higher willingness to pay [55]. The authors therefore suggested bundling only for the latter modes, whereas bike sharing and taxi services should be offered separately to increase the profitability of a public transport system.

Ho et al. [65] showed that respondents were willing to pay an average of GBP 5.28 for one hour of use in a free-floating car sharing model (start location unequal to destination) and GBP 4.32 for one hour of use of stationary car sharing (start location equal to destination) in a study conducted in Tyneside, UK [65]. On average, respondents were prepared to pay GBP 3.72 for one day of unlimited use of public transport. In general, studies show a

relatively low willingness to pay for MaaS, in some cases even below the cost of the service on the part of the providers [98].

The question of which MaaS offer is preferred depends mainly on the current use of mobility offers by users. The greatest potential for using MaaS was observed in the group that regularly uses public transport, as will as private vehicles [120]. Although there was little or no willingness to pay for the use of app and bike sharing, an MaaS app nevertheless provides incentives to use alternative modes. In their study conducted in Finland, Liljamo et al. [83] estimated an average willingness to pay of EUR 140, whereas the willingness to pay for MaaS was on average only 64% of the respondents' respective mobility costs [83]. The approval rates for MaaS were very high overall. In their study, 42% of the sample said they would like to have a monthly payment to cover all mobility expenses, whereas 81% supported the possibility of paying for multimodal trips with one ticket.

Hawkins and Habib [58] analyzed the value of mobility via surveys using the model of discrete choice in the Toronto and Hamilton areas (Canada) and added the criterion of geography [58]. In areas with low population density and few mobility alternatives, households were prepared to pay considerably more for an additional trip. The combination of car dependency and a high willingness to pay made these areas particularly attractive for private MaaS operators in terms of maximizing revenues.

### 5.4. Design of the Application Software

An MaaS app can be used for the planning, reservation, and booking of trips and the provision of real-time information, navigation, and travel assistance, as well as providing access to various means of transport, including comprehensive billing, and can be controlled via the customer's personal device. The growing variety of functions involves significant challenges related to the development of a simple and user-friendly design [107,115]. The main feature of an app is an easy comparison of different travel alternatives. This is realized by providing information on travel time, costs, possible connections, and ecological aspects [115].

Various studies have already been published regarding the design of the user interface, whether as a mobility assistance system [62] or map-based web platform [25,88]. Ultimately, an MaaS app, in addition to all the functions offered, should be intuitive to use and operate, and should be designed to attract the attention of potential users and bind them to the service in the long term. In addition to simple and understandable navigation and booking, the app should offer options for cancellation or plan changes [16].

### 6. Data and Technology

#### 6.1. Technical Realization

Another issue of great importance to a (new) mobility platform is its alignment with existing authorities and institutions, including political will, laws and regulations, formal institutions, and the willingness of travellers to place their trust in a platform [121]. Building a mobility platform in the area of MaaS requires many resources, whereby many functions are only made possible by technical knowledge. Beutel et al. [38] identified three main functions that characterize an MaaS system: (1) the collection of all necessary information, (2) the provision of functional components for administration, and (3) the provision of all services for use [38].

The standardization of data between transport companies and data providers, known as interoperability, and the unwillingness of transport companies to exchange data were identified by Polydoropoulou et al. [104] as obstacles to interconnection [104]. To overcome these hurdles, Marchetta et al. [88] emphasized interoperability between different technologies through a universal programming interface [88]. This enables devices, sensors, and users to access mobility services via a platform and, through the use of cloud services, databases can be separated from the actual business model and made accessible more easily [28].

A modular platform architecture for the simplified extension of functions can be enabled by the use of micro-services [39]. In this platform design, a service-oriented architecture is generated in which standardized micro-services efficiently and flexibly combine heterogeneous data sources such as real-time data and traffic options to provide end users with a customized mobility solution [92]. Separate micro-services allow transport operators to include external functionalities in their own services [39].

The MaaS platform provides an interface between users and various transport services, enabling communication between both sides. This requires technical developments: ICT on the transport services side and the necessary equipment in the form of a smartphone, app, and/or smart card on the user side [72]. In addition, the providers of MaaS must be enabled to quickly and reliably present alternatives in the highly complex decision-making system of their own and private offers, which a large number of existing local transport operators are currently unable to provide [101]. The complete integration of all functions within the app would make a smart card, including the associated infrastructure (e.g., smart card readers), unnecessary. The implementation of MaaS via smart cards is probably easier to realize in the short term and this can be replaced by smartphones and apps when technical possibilities develop further [72]. Extended to a block-chain-based smart contract, new types of automatic distributions of values and information between different stakeholders in the MaaS ecosystem can be enabled [74].

### 6.2. Data Management

The amount of data that are generated by mobility services is enormous. By utilizing this data, traffic volumes can be measured and predicted to improve traffic routing, optimize transfers, direct traffic, and much more [102]. In terms of MaaS, the generated data basis can be of interest for companies and organizations when combining gamification and MaaS. Via incentives integrated into the app for the use of certain transport services, valuable information on travel behavior, as well as on the preferences and habits of customers (e.g., in the use of the apps), could be collected [35]. The provision of real-time passenger information for certain transport services is also an opportunity to improve the customer experience and the connection between modes of transport [10].

In addition, traffic data offer further possibilities in the context of using an app. In the event of a disruption in operations, customers can be presented with reliable, real-time information that gives them options for further travel [115]. Furthermore, extensive travel data allow direct and personalized marketing with different MaaS user groups as targets. Thus, real-time data about trips, such as usage time, can be used to generate new revenue streams and for targeted advertising by external companies [32,44]. An outlook with regard to data analysis is provided by Wu et al. [127], who dealt with the consolidation of heterogeneous data from different sources with regard to MaaS [127]. To predict future travel patterns, the past travel behavior of users must be recorded. Based on this, it is possible to develop models that can also influence user decisions regarding travel through predictions.

### 6.3. Data Security and Resilience

An important challenge in the context of the safety of intelligent mobility is how information about citizens, traffic, and the city is handled. The protection, confidentiality, and privacy of all personal data must be guaranteed without compromising the operational efficiency of the system [32,59]. It is also necessary to clarify which actor in an MaaS ecosystem has the right to own and control the data. Therefore, policy makers should define standards for data collection, management, and dissemination to support the interoperability of data and increase the effectiveness of collaboration between MaaS stakeholders [73,105].

In a survey undertaken by Polydoropoulou et al. [104], urgent concerns about the disclosure of personal data via mobile MaaS applications were raised. Specifically, the chance of being tracked on a smartphone was worrisome for the interviewees, and they considered the control of the data collection process necessary [104]. The need to include

data protection and data management in the design of MaaS at an early stage is emphasized; otherwise, the requirements of the General Data Protection Regulation cannot be met [45]. Internal and external threads, which must be prevented with suitable measures, are also relevant to the safety of the overall system [40,128].

The release of data for public access is a driver of relevant and open-ended research on MaaS. To support the development of MaaS, the Finnish Transport Authority made all the data it produced freely accessible and established measures to support the use of open data [119]. An important factor is that the interoperability of the IT system has been made an important selection criterion in public procurement, whereas interoperability enables all devices, systems, and infrastructures to communicate information by reading, understanding, translating, and using each other's data [73,119].

Similarly, Russ and Tausz [108] highlight interoperability in MaaS development, since free access to data is a prerequisite for a service that is oriented towards the mobility needs of users. Interoperability of data facilitates cooperation at the regional, national, and international levels and provides the basis for an exchange of information in all directions [108].

## 7. Transformation of the Transport Ecosystem

### 7.1. Urban and Rural Areas

Due to continuing urbanization, the demand for mobility in cities is increasing steadily. The question of whether MaaS will reduce the absolute volume of traffic is still to be clarified and depends on the amount of people who use MaaS [10]. If bus trips are replaced by taxi or ride hailing rides, the total number of trips could rise because of the empty rides that occur when a vehicle is moving between two customers [79].

The data generated in a city from the regional transport and traffic network are currently only collected and used by public authorities. However, such data could also help external providers, such as developers and companies, to develop new urban-specific mobility solutions [102].

Arias-Molinares and Garcia-Palomares [29] conducted a case study in Madrid, Spain, analyzing the conditions for an MaaS offer in the city [29]. The authors pointed out two main conditions for the successful implementation of MaaS, namely, (1) a good working public transport system and (2) a variation of shared mobility services. Both criteria were met by the city, but still no working MaaS was detected. The reasons for this were an atomization of services, leading to more than 30 operators of shared and mostly electrified mobility services, and a lack of governance and collaboration in the development of MaaS.

Rural mobility is characterized by long distances and low population density, which results in a low demand for mobility and poor public transport connections. The provision of comprehensive mobility services in corresponding areas and the development and maintenance of the infrastructure pose great challenges for the dissemination of MaaS in such areas [48]. Regional and inter-regional cooperation can facilitate joint planning and procurement of mobility services and the exchange of information on best practices. In this way, the lack of resources can be counteracted in areas with a less developed infrastructure, which sometimes leads to incompatibility and a lack of flexibility in procurement and operation [48]. Using ICT for on-demand services can be a useful supplement to an exisiting MaaS scheme and public transport service to connect urban and rural areas [31]. The limited mobility market in rural areas leads to a lack of supply and competition, resulting in limited choices for users. Subsidies are an essential part of the mobility system in rural areas. If the introduction of MaaS is not supported by subsidies, private providers will demand higher prices and focus on the most profitable regions, which would not lead to an increase in overall welfare [48]. However, it should be noted that the scope of services does not necessarily have to be the same in all regions. Depending on the local situation, different subsidies and permits can be introduced by the public authorities, since rural areas are not exposed to the same mobility requirements as areas close to a community center [48].

*7.2. Impact on Modes of Transport and Environment*

7.2.1. Public Transport Providers

Instead of replacing existing systems, MaaS is commonly referred to as an instrument to solve the first-/last-mile problem in the literature [12]. With MaaS, gaps in current services due to fixed timetables or defined stops in public transport can be closed. As MaaS becomes more prevalent, the boundaries between public and private transport services become less separable, since public transport will be supplemented by new mobility options [10]. However, MaaS will not replace high-quality scheduled transport as the most efficient mode of transporting people in densely populated areas [6].

Public transportation is commonly seen as the backbone of an MaaS offering, which is reflected in a variety of publications [3,7,10]. The introduction of MaaS has the potential to trigger the increased use of traditional public transport, especially by reducing the use of private cars [6]. Thus, the potential customer pool of traditional public transport increases and generates higher revenues for public transport operators [6]. Wright et al. [126] discussed the integration of car pooling and MaaS to facilitate the access to public transport, which has shown promising results for areas with little access to public transport [126].

In contrast, Hörcher and Graham [66] drew attention to the possible negative effects of an increased market penetration of MaaS by modeling of the effects of an increased use of public transport and mobility alternatives [66]. An increased use of local public transport and the abandonment of private cars by some users could possibly lead to overcrowded buses and trains, especially if these tickets are sold as part of a fixed monthly subscription and the marginal cost of an additional trip for the users thus falls to zero. In turn, this would drive occasional users of public transport away from the offer, and they would instead cover more kilometers with their private vehicles. Smith et al. [6] observed a similar cannibalizing effect on public transport services exerted by MaaS through the provision of easier access to alternative modes of transport [6].

Wittstock and Teuteberg [124] noted the possibility of creating collective value through the realization of MaaS by providing comprehensive sustainable mobility. Many publications have assumed the existence of possible positive ecological effects of MaaS, although the actual impact on sustainability is still quite undefined. The influence of MaaS on greenhouse gas emissions within a certain region is also debatable, since different effects can steer total emissions in different directions [124]. Becker et al. [34] provided results of simulations that indicated the occurrence of reductions in green house emissions when a certain fleet size of car sharing and bike sharing was achieved. Up to that point, underused fleets can have negative impacts on emissions when they replace more sustainable ones [10].

Lyons [87] discussed the possibility of including increased walking as an adequate means of transport, especially to cover short distances [87]. Here, there are opportunities to offer incentives for the use of active transport modes via incentives.

7.2.2. Use of Private Cars vs. Sharing Services

MaaS aims to reduce the usage of private cars [6,16]. If designed correctly, the introduction of MaaS could result in a shift towards the use of transport services instead of private vehicles, especially for commuting and repetitive trips [10,117]. However, MaaS can also lead to a switch from public transport to less sustainable modes of transport such as car sharing or ride sharing, so observing the magnitude of both effects is important in order to be able to make an assessment of the sustainability of MaaS [27,43].

The UbiGo pilot project in Sweden was already able to prove this effect, the reasons for which were a general overestimation of one's own mobility needs, as well as the realization by the participants that other possibilities for covering their routes were available [75,118]. However, there are characteristics in user behavior that primarily point to a replacement of the second car. Despite good access to MaaS, many households keep their own first car, as this allows them to cover more unusual routes flexibly [65,117]. In this case, therefore, MaaS should rather be marketed as a replacement for the second car in a household or

MaaS should rather be regarded as a complementary service [117]. Simply offering an MaaS platform is not enough to achieve significant changes in user behavior [61,131].

Car sharing, both with electric and conventional vehicles, can be a strong catalyst for the use of MaaS and public transport if the service is easily accessible to users and, in the best case, is based on a free-floating concept [2,11]. In combination with these findings, the literature thus shows two competing effects, which overlap. The question of which of the two effects outweighs the other has not yet been conclusively clarified and remains to be determined in future work.

Bike sharing offers an alternative to car sharing, especially for short distances in urban regions. Although small car sharing and ride sharing fleets increase the demand for bike sharing, competition with large car sharing fleets reduces the demand for bike services [34]. In contrast, the demand for ride-hailing services seems to be independent of competition from other transport services. The reason for this is that car sharing and bike sharing compete for similar demand regions (both mostly serve central districts), whereas ride hailing providers serve a different segment of demand. This service connects districts, primarily the center and the outskirts of the city, so that car/bike sharing and ride-hailing are considered to be complementary services [2].

## 8. Research Gaps

Although the regulatory design of MaaS offerings with layered models of integration is already very advanced, the final design of the business model and the role of public authorities remain to be clarified. A clear regulatory framework that opens up a long-term perspective for investors and players and clearly allocates roles has been cited as a necessary prerequisite for the successful implementation of MaaS. However, it is still not certain how these will be designed in detail and whether regulations will have to be made at the local, regional, or national level.

The same applies to the role of public transport. Its importance for the concept of MaaS has been repeatedly emphasized; without public transport, no functioning MaaS system can be established. However, questions remain regarding how and with which roles the partnerships between local public transport and private mobility providers are to be designed in order to fulfill the goals of the local public transport on the one hand and the goals of private providers on the other.

In the user segment, there is a clear picture of the possible target groups, which are relatively young and educated users with a regular income. Segmentation of users into groups has been proposed as a solution for reducing the complexity of individualized offers, but there is still room for further work here. Additional questions have arisen with regard to individual users' and user groups' willingness to pay for these services.

In the data segment, the main questions that remain unanswered relate to sovereignty over the data and the type and scope of any access by mobility providers. Although private providers see many opportunities for monetarization in the use of data, data protection remains extremely important for users and public transport operators. The question of how solutions can be designed here that can anonymously increase the functionality of MaaS offers should be addressed in further work.

The effects on the transportation system in its current form have only been examined in rudimentary form, and there is room for further work here, especially in rural areas. The questions of how traffic flows will develop with the more widespread use of MaaS and, if necessary, how traffic routes will have to be adapted, must be studied more intensively, especially in simulations. Only by looking at changing traffic flows will it be possible to assess the environmental impact of MaaS, which has so far relied too heavily on the evaluation of individual potential than on the complex overall system.

In addition, various modes of transportation have been examined in detail, but there is still potential in terms of the impact and incentivization of walking as a mode of transportation. Conceivable approaches here would be to promote healthy behavior through credits or the avoidance of journeys, for example, in the case of overcrowding.

## 9. Conclusions

To summarize the findings of this literature review, the research questions posed at the outset are addressed below. The first and second research objectives were to identify the main areas of research on MaaS and to summarize the findings and results to date in due brevity. Among the 127 articles and publications included in the study, the main topics covered were the market, users, data and technology, and transformation of the existing transport system. In addition, the level of implementation of MaaS in the articles under review was recorded. The number of scientific publications about MaaS has grown rapidly in the past years. The variety of issues presented and analyzed in this review thereby reflects how broad the topics covered in the publications are.

The MaaS ecosystem requires the cooperation of various players at different levels; in this context, similar multi-level models for cooperation have been proposed. A coordinated approach that takes regional conditions into account and follows a uniform and coordinated strategy is essential. Public authorities have especially important tasks in the area of coordination and financing in order to ensure a balance between the interests of customers and providers.

For potential users of MaaS, previous mobility essentially determines which requirements an MaaS system must meet in order to be considered an adequate alternative. In terms of reliability, comfort, and flexibility, comparable characteristics must therefore be achieved with private vehicles, so that MaaS services can reach their self-set goals for reducing private car traffic. So far, potential user groups have been identified as young, progressive, and well educated-people, which limits the reach of MaaS to a subset of the population. Studies to date have revealed only a very low willingness to pay on the part of customers for MaaS, which consequently suggests that people have evaluated the concept without accounting for the possible abandonment of their own vehicles. The key to reaching a large number of customers is seen as the individualization of the offering, enabling customers to put together mobility packages tailored to their own needs. In addition to everyday individual mobility, tourist and work-related mobility can also be possible target areas for the development of MaaS.

The availability of data and information from a variety of sources forms the basis for a functioning MaaS system. The basic prerequisite for MaaS is therefore data from users and available transport options that are available and can be processed in real time, which places great demands on the regional infrastructure and the players involved. In addition, the high sensitivity of the data must ensure that dangers from internal and external threats are minimized as much as possible.

In the area of the transformation of existing transport systems, research has so far focused on urban areas and associated planning changes. Although some studies have focused on public transport in rural areas [136,137], this topic requires further research. More widespread use of MaaS would require additional transfer points in the form of mobility hubs, which would have a direct impact on urban planning in addition to transport planning.

Among all the articles examined so far, it is noticeable that, also due to a small number of practical applications, it is mainly challenges in the implementation of MaaS that have been named so far. Specific measures were the second most frequently mentioned, but impacts on existing systems and users were rarely mentioned.

A limiting factor of this work was the selection and analysis of the available literature. This is related to the databases that were used for the selection of the articles and which were available. In addition, MaaS is an enormously quickly developing research area, which is subject to permanent development. Therefore, with the system presented here we cannot claim to have collected and analyzed all the available literature, but the presented framework offers a basic framework for future analyses, which can be used for the further categorization of works.

**Funding:** This research received no external funding.

**Institutional Review Board Statement:** Not applicable.

**Informed Consent Statement:** Not applicable.

**Data Availability Statement:** Not applicable.

**Conflicts of Interest:** The author declare no conflict of interest.

## Abbreviations

The following abbreviations are used in this manuscript:

| | |
|---|---|
| DAP | Dynamic adaptive planning; |
| ICT | Information and computer technology; |
| IMI | Intermediary maas integrators; |
| IPT | Integrated public transport; |
| MaaS | Mobility as a service; and |
| TSP | Transport service providers. |

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
