# Peer review of "Literature Review of Mobility as a Service"

_sustainability, doi:10.3390/su14148962_

Round 1

Reviewer 1 Report

Dear author,

Please find in the attached file my comments to your manuscript.

Thank you.

Reviewer 2 Report

This paper presents a review on Mobility as a Service aims to highlight key research areas and further potential. The proposed study was conducted on June 15, 2020, but the topic is in continuous development, therefore other important contribution are needed, such as: “Models for Supporting Mobility as a Service (MaaS) Design”; “Mobility-as-a-Service as a transport demand management tool: A case study among employees in the Netherlands”; “Determinants of success and constraints of integrated ridesharing in rural areas”; “MaaS bundle design and implementation: Lessons from the Sydney MaaS trial”; “Exploring the MaaS market with systematic analysis” and others.

It would be appropriate to update the works at least to the first half of 2021.

In addition, the paragraph called “cooperation between market players” deals with customer involvement as an important component of the development of MaaS, but the methodologies for applying it are not considered, such as incentive mechanism (“Electric vehicle fleet relocation management for sharing systems based on incentive mechanism“), motivational approaches to involving people in new mobility services (“Shifting consumers into gear: car sharing services in urban areas”), and acceptance model based on important factors for users (“Factors that influence the acceptance of future shared automated vehicles – A focus group study with United Kingdom drivers”).

It was not easy to carry out the revision since the text was not in the format required by the journal and there was no line numbering. However, it would be advisable to revise the text to streamline it in the reading that appears heavy and long. English sentences need to be improved and typos be corrected (i.e. page 16, line 6, “to providing???”; pag. 17, line 21, “her” but the paper has more than one author….; after the colon there is always a capital letter….; and so on).

Furthermore, the references are to be thoroughly reviewed as it is not consistent with the indications of the journal; the numbering is not increasing but scattered, the names of authors greater than two should be indicated with “first author et al.” and not with everyone’s name.

Reviewer 3 Report

Page 1, Keywords: I understand that it is not easy to include all the basic concepts of MaaS in the Keywords. However, I would suggest you add Willingness-to-Pay (WTP), Public Transport, Users needs, Data management, Transport ecosystem.

Please follow Instructions for Authors (https://www.mdpi.com/journal/sustainability/instructions)

References must appear in numerical order within the manuscript. Please correct accordingly.

Please note that references 2,21,25,31,33,38,54,57,70,71,80 and 111 are missing from the text although they appear in the Reference List at the end of the paper.

Section 3 Overview of the MaaS literature: My suggestion is to include a map presenting the geographical distribution of the MaaS related research among the different countries and continents.

Page 5, last paragraph: My suggestion is to provide an explanation (within the manuscript) concerning the reason(s) for which “…most of the publications were conducted in Europe.”.

Table 2: Overview of MaaS literature: Please include the reference number ([  ]) in the 1st column. Please also include the names of all the coauthors (instead of “et al. “).

Page 13: Please do not use footnotes.

Section 5 MaaS Users: My suggestion is to try to present the information included in the specific Section using graphical representations (e.g., charts, histograms etc.) for the benefit of the reader.

Section 7.4 Environment: It is a rather small subsection. My suggestion is to merge 7.4 with 7.1 and 7.2.

Reference List: Please include the names of all the coauthors (instead of “et al. “). Please see, for example, references 4, 11, 13, 14, 15, 19, 20, 21, 24 etc.

Round 2

Reviewer 1 Report

Dear author,

Please find in the attached file my comments to this revised version of your manuscript. Thank you.

Reviewer 2 Report

The improvements carried out are appreciable for the purpose of an adequate quality level of the paper. However, the aspects highlighted in the first review and not yet developed in the document are significant in the context of the research conducted. It is therefore believed that such discussions will need to be present in order to improve the document.

Round 3

Reviewer 1 Report

Dear author,

Please find in the attached file my comments to this revised version of your manuscript. Thank you.

Reviewer 2 Report

The document has been improved and the adoption of 30 June 2021 as cut-off date is better suited to a continuously developing topic, even if the methodology used is not exactly appropriate. In fact, according to Fig. 1, the review update is carried out adding 19 papers to those previously identified, instead of starting from the first stage of the research. Also passing this approach, considering the improvement due to the updating of the research period adopted, the lack of discussion on the issue of customers involvement in MaaS services is significant. In fact, the concept of MaaS is based on the use of transport by users in a different way than in the past thanks to new technologies and new business models. To think that this approach works without customers involvement is incorrect, as they are the ones who ensure the success of this new idea for people’s mobility.
